# Exploring the association between stress-related hormonal changes, behaviours and facial movements after an interval training exercise in French Standardbred

Noémie Hennes[1,2]*, Léa Tutin[1], Aline Foury[3], Sylvie Vancassel[3], Hélène Bourguignon[2], Arnaud Duluard[4], Alice Ruet[1,5], Léa Lansade[1]

1 INRAE, CNRS, Université de Tours, PRC, Nouzilly, France, 2 French Federation for Horse Racing, Paris, France, 3 Univ. Bordeaux, INRAE, Bordeaux INP, NutriNeuro, UMR 1286, Bordeaux, France, 4 Société d'Encouragement à l'Elevage du Trotteur Français, Paris, France, 5 IFCE, 4940, Saumur, France

* hennes.noemie@gmail.com

## Abstract

Physical exercise can act as a physiological and a mental stressor. Monitoring exercise-induced stress is therefore essential to understand racehorses 'responses to effort and to ensure their welfare. Stress perceived by the horse during physical activity can be measured using various indicators, including stress-related hormones such as cortisol and adrenaline, and other neuromodulators such as serotonin, all involved in the stress response and its regulation. Another approach to assess physiological and emotional responses to stimuli such as exercise is through behaviours and facial movements. In this study, we aimed to 1) evaluate the changes in these three hormones following a trotting exercise, 2) determine the changes in behaviour and facial movements in response to the same exercise and 3) investigate potential relationships between hormonal variations and specific behavioural patterns that could serve as indicators of exercise-induced stress in horses. Fourteen French Standardbred horses from two stables were monitored over one day. In the morning, they performed an interval training trotting exercise. Behaviours and facial movements were recorded via video for 2 min 30 both before and just after exercise. Saliva and blood samples were collected at four time points: before exercise, just after exercise, 1 h post-exercise and 24 h post-exercise to assess salivary cortisol, and serum concentrations concentration of adrenaline and serotonin. Results showed significant post-exercise increases in all three hormones with peak concentrations observed immediately after exercise, and elevated cortisol and adrenaline levels persisting one hour later. These variations are consistent with normal physiological responses to physical effort, reflecting activation of regulatory systems rather than necessarily indicating negative stress. However, inter-individual variability in the magnitude of these responses suggests that horses did not all experience the exercise in the same

**Data availability statement:** The code and dataset underlying the results presented in this study are available from the following public repository: https://entrepot.recherche.data.gouv.fr/dataset.xhtml?persistentId=doi:10.57745/QGZQ5J&faces-redirect=true.

**Funding:** This study was funded by the French National Institute for Horse and Riding (CSIFCE2022), the French Association for Research and Technology (ANRT), and the French Horse Racing Federation through a CIFRE fellowship (contract no. 2022/0468). The funders had no role in study design, data collection and analysis, decision to publish, or preparation of the manuscript.

**Competing interests:** The authors have declared that no competing interests exist.

way, highlighting potential differences exercise-induced stress. In terms of behaviour, horses exhibited higher frequencies of facial movements, particularly mouth movements, after exercise than before. Moreover, increases in serotonin and adrenaline concentrations were positively associated with agitation-related behaviours (pawing and head turning) and mouth movements. Overall, our findings suggest that a behavioural profile characterised by increased agitation and mouth movements may reflect a post-exercise arousal response in French Standardbreds. These behaviours, in association with hormonal changes, could provide a useful non-invasive tool to assess to assess horses' response to exercise, and potentially exercise-induced stress. However, further studies are needed to confirm this interpretation, particularly by investigating the potential effects of post-exercise management practices such as cross-tying, which may induce frustration.

## Introduction

For athletes, including racehorses, physical training is a necessary but potentially demanding part of preparing for competition. Monitoring how individuals respond to exercise, both physiologically and psychologically, is therefore essential to optimise both performance and welfare. High-intensity physical exercise is considered both a physiological and psychological stressor [1,2], referred to in this paper as exercise-induced stress. To restore homeostasis, the body activates different regulatory systems, notably through hormonal responses [3,4]. Acute stress responses involve two major systems: the Sympathetic-Adrenal-Medullary (SAM) axis and the Hypothalamic-Pituitary-Adrenal (HPA) axis [3]. The SAM system triggers the rapid release of adrenaline, which prepares the body for immediate action by increasing heart rate, blood pressure, and energy availability. Meanwhile, the HPA axis releases cortisol, a hormone essential for energy metabolism and longer-term stress regulation [3]. During recovery, these hormones act on the brain to restore homeostasis by down-regulating the stress response. Otherwise, chronic elevated levels of those hormones can be detrimental to the organism, compromising welfare [5,6]. The magnitude of these hormonal responses appears to be dependant of the intensity and duration, with more intense efforts leading to greater elevations in both cortisol [7] and adrenaline [8] and inter-individual differences in these responses could be interpreted as reflecting differences in perceived effort or experienced stress.

In both horses and humans, elevated concentrations of cortisol have been observed in response to endurance exercise in humans [9] and horses [10], racing [11] or lunging with constrained neck positions [12]. Cortisol increase can be measured in plasma or saliva, the latter being preferred for its ability to avoid the confounding effects of blood collection [13,14]. Although less studied in horses, adrenaline concentration in serum has also been shown to be affected by exercise [15,16], such as heavy resistance exercise in humans [17]. Serum serotonin (5-HT) is another neurochemical involved in stress regulation and homeostasis recovery [18]. It plays roles in regulation of gastrointestinal function, modulation of the blood vessel

tone [19], and the mood regulation [20]. 5-HT is also implied in responses to inflammation and pain perception, with both pro- and anti-nociceptive effects depending on the situation [21–23]. During exercise, which involves moderate muscular inflammation, it is more likely to exert an anti-nociceptive effect, hypothesis that is strengthen by findings that a lack of 5-HT hampers physical performance in rats [24]. Increases in serotonin concentration have been observed after aerobic exercise in humans [25,26] or treadmill and simulated race in horses [1,27].

In addition to hormonal changes, behaviours and facial movements can reflect horse's responses to stressful situations including exercise. For example, Trindade and colleagues [28] identified weight shifting as an indicator of fatigue in ranch horses, correlating with physiological fatigue parameters. Moreover, higher discomfort behaviours during exercise have been found to be linked with higher rest time after exercise in French Standardbreds, suggesting a link between behaviours and recovery [29]. Facial expressions have also emerged as promising indicators of stress, pain and discomfort. In horses, specific facial expressions and movements have been identified to evaluate pain [30–33], and similar findings have been reported in other species, including cats [34,35] and rodents [36,37]. Additionally, positive anticipation and frustration, have also been linked to specific combinations of facial movements and behaviours [38,39], suggesting that behavioural and facial movements cues can be used to assess horses' physical pain, responses to stressors, but also broader emotional states in horses.

In this study, we investigate the hormonal changes induced by trotting exercise in French Standardbreds and explore associated behaviours and facial movements. The objective is to identify post-exercise behaviours and facial movements that may serve as accessible, non-invasive indicators of horses' responses to exercise, with potential applications in training management and welfare monitoring. We hypothesise that salivary cortisol, adrenaline, and serotonin concentrations increase after exercise compared to before, and that behavioural and facial movements also change in response to exercise. Furthermore, based on the assumption that greater hormonal increases reflect higher levels of exercise-induced stress, we expect to find associations between hormonal changes and specific behaviours or facial movements, which could then be proposed as behavioural indicators of elevated exercise-induced responses, including stress.

## Materials and methods

### Animals

The study was conducted on fourteen French standardbreds (four geldings, six mares and four stallions) aged between three and one year old (mean±SD: 5.07±1.90), from two training stables located in France. The subjects were maintained under similar conditions, with the majority of their time spent in single stalls measuring 3.5 x 4m or 4x4m. The stalls were bedded with straw, and the animals were provided with hay distributed twice a day and additional concentrated food tailored to the individual based on their age, training status, and physical condition. They had access to *ad libitum* water and were turned out in pasture or paddock for 2–4 hours per turnout, at least twice a week.

The horses were race-fit, as they all had either recently raced or were scheduled to race in the near weeks. They were trained for harness racing by professional equine trainers. In terms of training, they underwent one to two sessions of intense exercise (interval training) per week, with the remaining days comprising light work (trotting), walking, or outings to the pasture or paddock. All training sessions were conducted between 8 a.m. and 1 p.m.

### Design of the study

Horses included in the project had not raced for at least four days prior to the start of the study. Data were recorded over two consecutive days for each stable in March 2024. On the first day of the study, the horses performed a trotting exercise. The exercise was carried out in the morning and was equivalent to interval training, in which horses performed a warm-up at a slow trot, followed by repeated accelerations (2–6 per horse, mean±SD: 3.10±1.19) interspersed with recovery phases at a moderate trot. Each acceleration lasted 4.15±2.36 min (total duration per horse: 9.29±1.83 min) and covered on average 2.54±1.60 km (total acceleration distance: 5.56±0.91 km). The recovery phases between

accelerations lasted 3.23±2.16 min on average (total duration: 6.72±3.45 min) and covered 0.76±0.52 km on average (total iner-accelerations distance: 1.73±1.30 km). After the last acceleration, horses cooled down at slow trot before going back to the stable. Overall, horses covered on average 11.5±1.55 km, equivalent to 33.0±5.60 minutes' exercise.

The horses were exercised in groups of two or four, with all horses of the stable divided into four consecutive groups from 6:30–13:30 a.m. During exercise, the horses wore a commercial heart rate belt (Polar Trotter Equine) with a POLAR captor (Polar hear rate monitor - H10), which allowed heart rate, distance and speed to be recorded using the Polar Beat application on smartphones held in the rider's pocket for the entire duration of the exercise. Due to a problem with the Polar Beat application or GPS data, one horse was not recorded for heart rate during exercise and four horses were not recorded for speed. On average, the horses had a mean heart rate during exercise of 145.0±9.1 bpm and a maximum heart rate of 217.5±10.2 bpm, with a mean speed of 21.8±1.3 km/h and a maximum speed of 47.2±3.7 km/h.

After exercise, the horses' equipment was immediately removed and they were showered with water, scraped to remove excess water, covered with a blanket and cross-tied in their stalls, as they usually are after exercise. Horses were not allowed to roll after exercise. Salivary and blood samples were collected four times: 30–60 minutes before exercise (from 6:15–9:30 am depending on the horse, horses who were exercised first were also collected first), just after exercise (post exercise), one-hour post-exercise and 24h post-exercise (Fig 1). Behavioural and facial movement recordings were recorded while horses were cross-tied, both before and 3–5 min post exercise (after showering), to ensure the consistency across conditions.

The day before exercise, a veterinary examination was conducted based on four inclusion criteria: heart rate, respiratory rate, dehydration and intestinal noises.

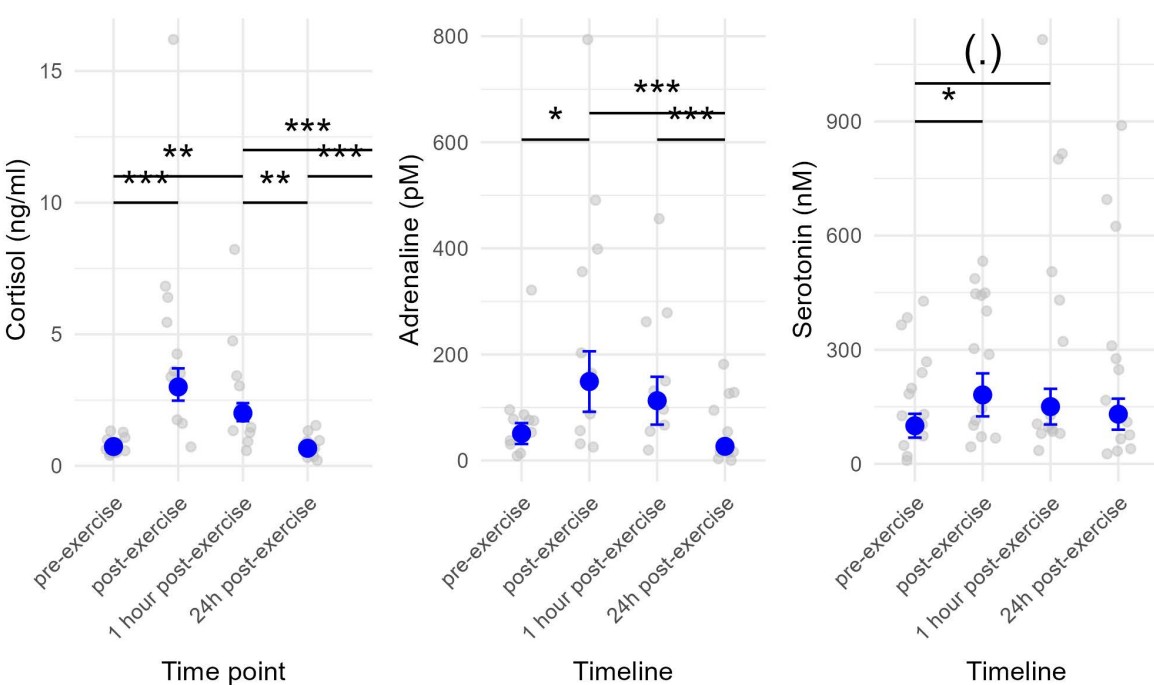

**Fig 1. Salivary cortisol, serum adrenaline and serum serotonin concentrations before and directly after (post-exercise), one hour after (1-hour post exercise) and 24h after (24h post-exercise) exercise.** Grey points represent raw data whereas blue points and lines represent respectively means and standard deviations estimated by the general linear model including time points as fixed effects and stable and horse as random effects for each hormone. The stars represent the significance of post-hoc analyses (emmeans). *: $p < 0.05$, **: $p < 0.01$, ***: $p < 0.001$, (.) $0.05 < p < 0.1$.

## Physiological data

**Salivary cortisol.** Saliva samples were obtained by inserting a saliva swab (SARDTEDT cortisol Salivette®) into the horse's mouth using a clamp. The experimenter then proceeded to pass the saliva swab over and under the tongue for approximately 30 seconds, ensuring that it was thoroughly soaked. The experimenter took care to place the clamp on the part of the mouth where the horse had no teeth to avoid any potential damage to the teeth. One horse was excluded from this procedure due to exhibiting stress-related behaviours (such as high-up head posture, head shaking, and other defensive behaviours). However, none of the other horses displayed such behaviours and remained calm throughout the experiment. Given that salivary cortisol is a marker of acute stress, salivary samples were collected prior to collecting blood samples, in order to avoid any potential modification of salivary cortisol levels due to stress related to the blood sampling process. Salivary samples were directly conserved in polystyrene boxes containing twenty kilograms of dry ice (−80°C) before being transferred in −80°C freezer and being analysed.

**Blood collection, adrenaline and serotonin.** Venepuncture of the jugular vein was employed to obtain blood samples of 9mL (sterile well tube – BD Vacutainer®). All horses were accustomed to such sampling procedures and none exhibited signs of distress or aggression during the process. To mitigate the risk of haematoma formation, blood samples were collected alternately from the right and left jugular veins. Following the collection of a blood sample, the horses were scratched at the withers by the individual responsible holding them as a form of positive reinforcement. Prior to the decanting process, the tubes were rotated three times to ensure thorough mixing of the sample. Following this, the tubes were left to decant for 45 minutes at ambient temperature. Subsequently, the samples were subjected to centrifugation for 12 minutes at 3250g using a portable centrifugal machine (Dutscher EBA 200 HETTICH). The resulting serum was divided into aliquots, placed in Eppendorf tubes, and stored in polystyrene boxes containing dry ice (−80°C) until the end of the day. At this point, the samples were transferred to −80°C freezers for subsequent analysis.

No anaesthesia or analgesia was required for saliva or blood collection.

## Dosages

**Salivary cortisol.** Cortisol was measured in 50 μl saliva sample using an enzyme immunoassay (Cortisol Saliva ELISA, IBL International GmbH, Hamburg, Germany). The intra-assay and inter-assay coefficients of variation were 4.3% and 13.2%, respectively. The assay sensitivity was 0.06 ng/ml.

**Serum adrenaline and serotonin.** Adrenaline and serotonin were extracted from 1.5 ml and 100 μl serum respectively (Catecholamines in Plasma – HPLC and Serotonin in Serum/Plasma/Whole Blood – HPLC respectively, Chromsystems Instruments & Chemicals GmbH, Munich, Germany). After extraction, samples were stored at −80 °C until use. Quantitative determination was performed by high-performance liquid chromatography (HPLC) coupled to electrochemical detection.

For adrenaline, 20μl of the sample were injected into an HPLC system equipped with an Equisil ODS (C18) HPLC column (150 x 4.6 mm, 5 μm; C.I.L.-Cluzeau) and a coulometric analytical cell (200 mV for the potential of oxidation; Cell 5011, ESA, Paris, France) connected to the coulometric detector (CoulochemII, Eisa, Paris). The flow rate of the mobile phase was 1.2 ml/min. The mobile phase was composed as follow: 60 mM $Na_2HPO_4$, 0.1 mM EDTA, 7% (V/V) methanol, 100 μL triethylamin, and sodium octyl sulfate (200 mg/L) dissolved in ultrapure water (18 MΩ.cm-2). A pH 6 was chosen to put the acidic metabolites in the solvent front. The detector was connected to a computer via an interface and the corresponding software (Ulyss, Azur system version 5, Alphamos, Toulouse, France). The system was calibrated using different concentrations of adrenaline while a standard solution was injected daily before the analysis of each serie of samples.

For serotonin, 20 μL of the sample were injected into an HPLC system equipped with a 5 μm C18, 3x100 mm silica column (ACE, AIT France, Cormeilles-en-Parisis, France) and coupled to an electrochemical detection system (Antec Decade 2, CJ Lab, La Frette, France). The flow rate of the pump allowing the circulation of the mobile phase (0.1 M sodium acetate, 0.1 M citric acid, 1 mM diethylamine, 1 mM sodium octyl sulfate and 0.1 mM EDTA) was set at 0.4 mL/min.

Levels were identified through their retention times, as compared to their respective standards injected daily and quantified using the Chromeleon integration 6.8 software (Dionex, Sunnyvale, CA, US).

## Behaviour and facial movements

The behaviours and facial movements of the horses were recorded for a period of two minutes and thirty seconds using two 4K video cameras (Sony FDR-AX43). If the horse was moving and thus, was not visible in the camera during the two minutes and thirty seconds, we extended the video recording duration up to three minutes. In order to record the horses' facial movements and ears positions, the cameras were positioned at a distance of 2m from the stall door, aligned with one side of the opening. This allowed for the horses' faces to be captured in a three-quarter view. In order to record the horses' behaviours, a second camera was positioned at a distance of 3m from the stall door, aligned with the opposite side of the opening to that of the camera used for facial movements recording. An observer was stationed behind each camera to adjust the positioning in the event that the horse moved and to secure the horse and the camera in the event of any problem. The two cameras were recording simultaneously for a period of two minutes and thirty seconds, capturing the subject's face and entire body. The horses were cross-tied in their stalls by means of two ropes or chains, with one side of the halter fastened to a corner of the stall or stall door, as is customary during the preparation and care of the animals.

Prior to coding, all video recordings of horse faces were watched once at normal speed. The recordings were then coded using an ethogram (Table 1), comprising Action Descriptors (AD) and Action Units (AU) from the Equine Facial Action Coding System (EquiFACs), as well as ears positions and other facial movements or behaviours that had been observed during the initial observation. Some of the Action Descriptors (AD) or Action Units (AU) were regrouped in accordance with their simultaneous appearance. Video segments in which the horse was not positioned at an optimal angle or in which the lighting was inadequate for discerning facial expressions were excluded from the analysis, resulting in a

**Table 1. Ethogram of the facial expressions from the equiFACs coding system [40] that have been regrouped (for facial movements that occurred simultaneously, or difficult to differentiate) and behaviours recorded from the video of horses 'face.**

| Facial movement | EquiFACs code | Description |
|---|---|---|
| *Ears* | | |
| Ears forward [d] | | Both ears are turned forward so their inner part is pointing forward |
| Ears midline [d] | | Inner parts of both ears are turned on the side. |
| *Eyes* | | |
| Blink and half blink [f] | AU145 and AU47 | The upper eyelid moves towards the lower eyelid to close the eye, either completely or partially. The lower eyelid can also be pulled up. |
| Upper eyelid lift and white eye increase [f] | AU5 and AD1 | The upper eyelid is pulled upwards, making the white of the eye more visible. |
| *Mouth* | | |
| Lip puller [f] | AU113 and AU12 | The lateral part of the lip is pulled obliquely or towards the ears. |
| ULT [f] | NA | Upper Lip Tremor: Lateral movement of the upper lip. |
| LLT [f] | NA | Lower Lip Tremor: Lateral movement of the lower lip. |
| Chain or rope chewing [d] | NA | The horse grabs the rope or chain to which it is attached and may chew on it. The facial movements of the nose and muzzle are not coded when this behaviour occurs. |
| Chewing [d] | NA | The horse is engaged in a chewing movement. During this movement, the facial movements of the muzzle and mouth are not coded. |

[f]: coded as frequency,

[d]: coded as duration.

mean duration of the coded videos of 150.21±1.14 seconds. Observers were validated in the use de EquiFACs coding system prior to code the videos.

All video recordings of the entire body of the horse were also viewed once at normal speed in order to construct the ethogram (Table 2). Subsequently, the videos were watched three additional times at a slower speed in order to code the horse's leg movements, head movements, and general behaviours.

The Boris Observer software was employed to code the videos using continuous recording method. Subsequently, the resulting data were extracted in order to obtain the frequencies and/or durations of each facial expression and behaviours.

**Intra and inter observer's reliability.** To assess the intra-reliability of the EquiFACS coding, six randomly selected videos (10% of the total) were re-coded by the same observer. The Cohen Kappa reproducibility test was employed to assess both forms of reproducibility. This test showed good intra-reliability for one facial action unit (AU17) and very good intra-reliability for 17 facial action units (see S1 Table in S1 File).

## Statistical analyses

Statistical analyses were performed using the Rstudio software version R.4.4.3 [41].

**Hormonal changes.** To assess the effect of time point on hormone concentrations, one generalised mixed model was constructed for each of the three hormones using the glmmTMB function from the package of the same name [42]. Time point was included as fixed effect factor while stable and horse were included as random effects factors. The significance of the time point effect was evaluated using an ANOVA (Anova function, car package [43]). Each model was then compared to its corresponding null model (i.e., random effects factors only) using an ANOVA to ensure it provided a better explanation of the data. To check the fit of the residuals, the DHARMA function from the DHARMA package [44] was used. Finally, differences between pairs of time points were checked using the post hoc test emmeans [45].

**Changes in behaviours and facial movements after intense exercise.** To construct the first pre and post-exercise behaviours Principal Component Analysis (PCA1) performed on the pre and post-exercise behaviours and facial movements, we excluded "Ears midline", AU27, LLT and "Chain or rope chewing" that showed more than 70% of collinearity with respectively "Ears forward", AU26, AU17 and AD160 or head turning. Moreover, as one horse showed extreme values in the frequencies of facial movements, we decided to remove this individual from the following analysis.

Table 2. Ethogram of the behaviour recorded from the videos of the entire body, adapted from [28].

| Behaviour | Description |
|---|---|
| Bodyshake [f] | The horse rotates its head, neck and upper body quickly and rhythmically along its long axis while keeping its feet planted. |
| Head turning [d,f] | The horse holds its head to one side (either left or right) with a flexed neck, focusing on a stimulus in its environment. One or both ears are directed forward. The horse stays in this position for a few seconds before returning its head to align with the longitudinal axis of its body. |
| Headshake [f] | The horse turns its head to the right or left, making the white of its eye visible. |
| Head tossing [f] | The horse moves its head gently up and down. |
| Locomotion [f] | The horse moves at least two limbs resulting in a change of position. |
| Paw [f] | The horse lifts one foreleg off the ground and extends it forward, then drags it backward with the hoof scraping the ground. |
| Hind limb lift [d,f] | The horse lifts the hind leg, flexes the tarsus, holds the lifted leg for at least 1 second and then lowers it back to the same position. |
| Resting leg [d] | The horse bears its weight on three legs, keeping one hind leg relaxed and unloaded. |
| Weight shifting [f] | The horse initially supports its weight on all four legs, then shifts to bearing weight on three legs, leaving one foreleg unloaded for at least 1 second. |

[f]: coded as frequency,

[d]: coded as duration.

The selected variables that showed a correlation of 0.5 or more with at least one of the two first principal components, as shown in S2 Table (S1 File), were used to construct the final pre and post exercise behaviours PCA (PCA2). PCAs were performed using the FactoMineR package [46].

Differences between the two groups (pre- vs. post-exercise) were then assessed by comparing the positions of the graphical barycenters on the first two PCA dimensions using a MANOVA.

**Interrelations between hormonal changes, behaviours and facial movements.** To construct the post-exercise PCA (PCA3), we excluded "Ears midline", AU27, LLT, AUH13, "Head tossing" and "Chain or rope chewing" that showed more than 70% of collinearity with respectively "Ears forward", AU27, AU17, "head turning", AD19 and chewing. Subsequently, only behaviours and facial movements that were associated to at least one of the two first dimensions of the PCA with a coefficient higher than 0.5 were retained to perform the final post exercise PCA, as presented in S3 Table (S1 File). Then individuals' coordinates on the first two dimensions of this PCA were extracted and used as response variables in general mixed models. One full model was built for the first principal component of the PCA (Dim1) and another for the second (Dim2) using the glmmTMB function. Salivary cortisol, adrenaline, and serotonin variations (post-exercise concentration minus pre-exercise concentration) were included as fixed effects, while stable was added as a random effect. Then, model selection was carried out using the dredge function from the MuMIn package [47], based on the Akaike Information Criterion (AIC), to identify the model best explaining the response variable distribution. Finally, an ANOVA was performed on the selected model to assess the significance of fixed effects factors and the fit of the residuals were checked using DHARMA package.

**Ethical statement.** This study was conducted in accordance with the European Directive 2010/63/EU and the French rural code (articles R.214-87 to R.214-126). The project was approved by the French Ministry of Higher Education and Research under authorization number APAFIS#47916–2024022115122964 v3, after evaluation by the Animal Experimentation Ethics Committee no. 019.

## Results

### Hormonal changes

**Salivary cortisol.** The model for salivary cortisol showed that the time point significantly influenced salivary cortisol concentration (Table 3) and accounted for 56.4% of the variability in the results. Specifically, cortisol levels were higher immediately after exercise and 1-hour post-exercise compared to pre-exercise levels (emmeans; pre-exercise vs. post-exercise: t-ratio = 7.809, p < 0.001; pre-exercise vs. 1-hour post-exercise: t-ratio = 4.665, p < 0.001). Salivary cortisol concentration returned to baseline 24 hours after exercise (emmeans; pre-exercise vs. 24h post-exercise: t-ratio = −0.344, p = 0.986).

**Adrenaline.** Time point also had a significant effect on serum adrenaline concentration (Table 3) but explained less the variability (marginal $R^2$ = 0.291), while random effects (i.e., horse and stable) played a more substantial role compared to the cortisol model (conditional $R^2$ = 0.549, with 25.8% attributed to random effects). Post-exercise adrenaline levels were higher than both pre-exercise and 24-hour post-exercise levels (emmeans; pre-exercise vs. post-exercise: t-ratio = −3.10,

Table 3. Summary of the model results on cortisol, adrenaline and serotonin with Time Point as a fixed effect. Model type refers to generalized linear models (GLM) with random effects for Horse and Stable.

| Hormone | Model type | Transformation | Chi² | Degree of freedom | p-value | Marginal R² | Conditional R² |
|---|---|---|---|---|---|---|---|
| *Cortisol* | glm | Inverse | 92.4 | 3 | 2.2e-16 | 0.564 | 0.694 |
| *Adrenalin* | glm | Log | 31.2 | 3 | 4.751e-07 | 0.291 | 0.549 |
| *Serotonin* | glm | Log | 11.2 | 3 | 0.0105 | 0.0413 | 0.797 |

p<0.05; post-exercise vs. 24-hour post-exercise: t-ratio=5.14, p<0.001). Serum adrenaline concentration returned to baseline 24 hours after exercise (emmeans; pre-exercise vs. 24-hour post-exercise: t-ratio=2.05, p=0.187).

**Serotonin (5-HT).** 5-HT serum concentration also appeared to depend on time point, but to a much lesser extent than the two other studied hormones. Indeed, time point explained only 4.13% of the variability in the results, whereas random effects were dominant (conditional $R^2$=0.797, with 75.6% attributed to random effects). More specifically, 5-HT levels were higher post-exercise than pre-exercise and tended to remain elevated 1-hour post-exercise (emmeans; pre-exercise vs. post-exercise: t-ratio=−2.99, p<0.05; pre-exercise vs. 1-hour post-exercise: t-ratio=−2.62, p=0.0549). Like salivary cortisol and adrenaline, 5-HT concentration 24-hours post-exercise did not differ from pre-exercise levels (emmeans; pre-exercise vs. 24-hour post-exercise: t-ratio=−1.24, p=0.604). More details of the models are presented in S4 Table (S1 File).

### Changes in behaviours and facial movements after intense exercise

The first dimension of pre and post-exercise PCA explained 28.3% of the variance and the second dimension explained 19.7%. The first dimension was mainly positively associated with lower lip depressor (AD16), tongue show (AD19), upper lip raiser (AU10), chin raiser (AU17), jaw drop (AU26), upper lip tremor (ULT) and ears forward. The second dimension was positively correlated with, blink or half blink (AU145+AU47), chewing, head turning, and hind limb lift, whereas it was negatively correlated with lip puller (AU113+AU12) and upper lip tremor (ULT) (Table 4).

A comparison of the graphical barycentre's of individuals (Fig 2) in the pre-exercise and post-exercise groups showed a significant difference in their placement (MANOVA; F=5.85, p<0.01), particularly along dimension 1 (ANOVA; Dimension 1, F=12.1, p<0.01). Specifically, horses exhibited lower lip depressor (AD16), tongue show (AD19), upper lip raiser (AU10), chin raiser (AU17) and jaw drop (AU27) more frequently post-exercise than pre-exercise and spent more time

**Table 4. Loadings of facial movements and behavioural variables in the final pre and post-exercise Principal Component Analysis (PCA2) for 13 French Standardbreds. Data from two time points (pre- and post-exercise) were used to perform the PCA. The table presents the loadings of each variable on the first two principal components (Dimension 1 and Dimension 2).**

|  | Dimension 1 | Dimension 2 |
|---|---|---|
| *Facial movements* | | |
| *Ears* | | |
| Ears forwards: EAD101 | **0.674** | −0.260 |
| *Eyes* | | |
| Blink and Half blink: AU145+AU47 | 0.348 | **0.561** |
| Upper eyelid lift and white eye increase: AU5+AD1 | 0.471 | 0.388 |
| *Mouth* | | |
| Upper lip raiser: AU10 | **0.650** | −0.365 |
| Lip puller: AU113+AU12 | 0.285 | −0.503 |
| Lower lip depressor: AU16 | **0.743** | −0.053 |
| Chin raiser: AU17 | **0.729** | −0.187 |
| Tongue show: AD19 | **0.608** | 0.416 |
| Jaw drop: AU26 | **0.651** | 0.375 |
| Upper Lip Tremor: ULT | **0.639** | **−0.523** |
| Chewing | 0.127 | **0.566** |
| *Behaviours* | | |
| Head turning | −0.085 | **0.625** |
| Hind limb lift | 0.303 | **0.574** |

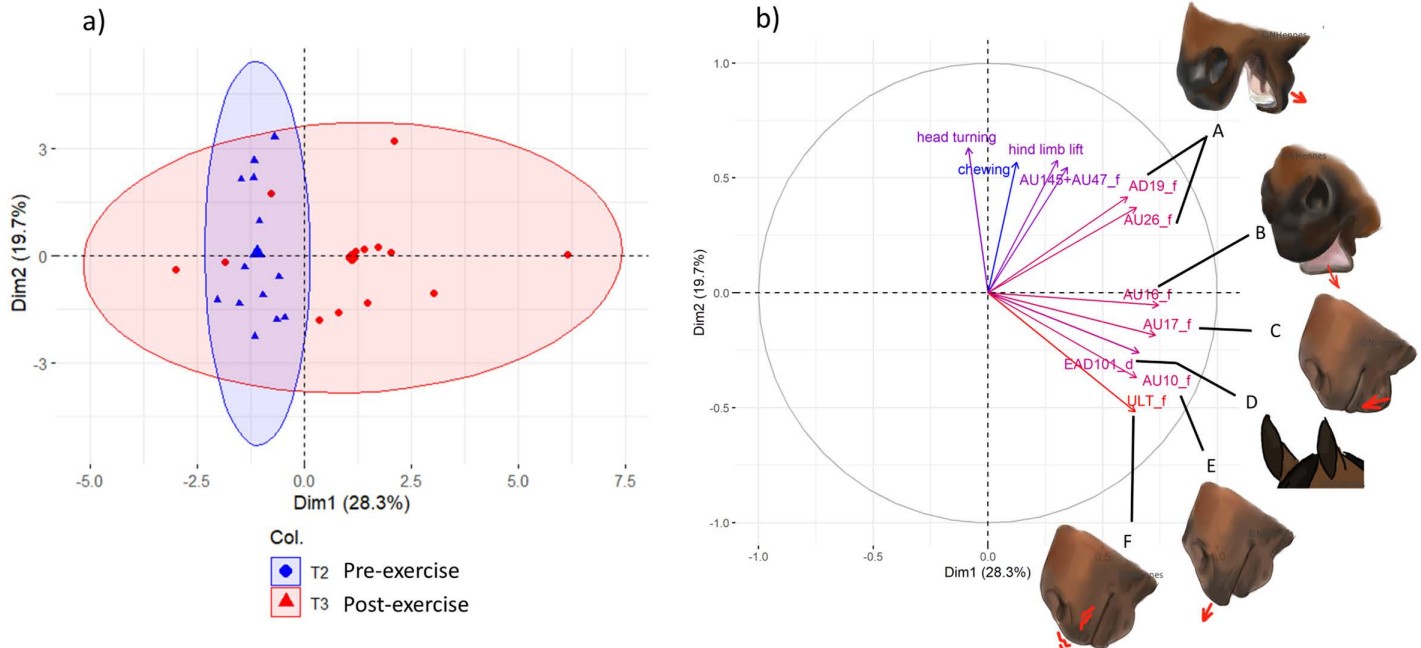

**Fig 2.** Principal component analysis of behavioural and facial expression changes pre- and post-exercise in horses. **a)** Projection of individuals onto the first two principal components of the PCA performed on behaviours and facial movements at two time points: pre-exercise and post-exercise. The largest point and triangle represent the barycentres of the individuals' distribution. **b)** Projection of the variables showing at least 50% correlation with one of the first two dimensions of the PCA. Drawings A to F correspond to facial movements positively correlated with the first principal component. The horses tend to express more often or longer (ears forwards) those facial movements post exercise compared to pre-exercise.

with their ears forward. However, no significant difference was observed in the positioning of the barycenters on the second dimension of the PCA (MANOVA; F = 0.04, p = 0.853).

### Interrelations between hormonal changes, behaviours and facial movements

The post-exercise PCA showed two main dimensions that respectively explained 25.4% and 23.6% of the variance. The first dimension was positively associated with tongue showing (AD19), upper lip raiser (AU10), jaw drop (AU26), upper lid raiser and eye white increase (AU5+AD1), ears forward, resting leg and negatively associated with pawing (Table 5). The second dimension was positively associated with lower lip depressor (AU16), sharp lip puller and lip corner puller (AU113+AU12), chin raiser (AU17), upper lip tremor (ULT), head turning, pawing and negatively correlated with weight shifting (Fig 3). In view of these correlations, we named the first axis "Calm vigilance" and the second axis "Agitation and mouth movements".

Then, based on the AIC comparison, the selected model construct from the full model for "Calm vigilance" was the null model and was the model that take adrenaline and serotonin as fixed effects for the dimension of the PCA named "Agitation and mouth movements" (Table 6). Details of this latest model and the results for the comparisons with the null model are presented in Table 7.

### Discussion

Our results were consistent with our hypotheses, as exercise was found to (1) increase salivary cortisol, serum adrenaline and serotonin concentrations up to 1-hour post-exercise (cortisol and adrenaline), (2) induce changes in behaviour and

**Table 5. Loadings of facial movements and behavioural variables in the final post-exercise Principal Component Analysis for 13 French Standardbreds. Data from post-exercise were used to perform the PCA. The table presents the loadings of each variable on the first two principal components (Dimension 1 and Dimension 2).**

| | Dimension 1 | Dimension 2 |
|---|---|---|
| *Facial movements* | | |
| *Ears* | | |
| Ears forwards: EAD101 | **0.610** | 0.405 |
| *Eyes* | | |
| Upper eyelid lift and white eye increase: AU5+AD1 | **0.760** | 0.109 |
| *Mouth and nostrils* | | |
| Upper lip raiser: AU10 | **0.742** | 0.106 |
| Lip puller: AU113+AU12 | 0.087 | **0.559** |
| Lower lip depressor: AU16 | 0.406 | **0.594** |
| Chin raiser: AU17 | 0.270 | **0.771** |
| Tongue show: AD19 | **0.796** | −0.122 |
| Jaw drop: AU26 | **0.769** | 0.068 |
| Upper Lip Tremor: ULT | 0.423 | **0.590** |
| Nostril dilatator: AD38 | 0.228 | −0.432 |
| *Behaviours* | | |
| Head turning | −0.199 | **0.641** |
| Headshake | 0.177 | −0.488 |
| Locomotion | 0.370 | −0.442 |
| Paw | **−0.612** | **0.532** |
| Resting leg | **0.501** | −0.421 |
| Weight shifting | 0.145 | **−0.687** |

facial movements, and (3) the greater the hormonal variation (pre vs post-exercise) in adrenaline and serotonin, the more likely specific facial signs and behaviours were observed, which could serve as indicators of higher responses to exercise, including stress. In particular, we observed a positive relationship between agitation-related behaviours (i.e., increased duration of head turning as well as higher frequency of pawing), mouth movements (i.e., lower lip depressor (AU16), sharp lip puller and lip corner puller (AU113+AU12), chin raiser (AU17), upper lip tremor (ULT)), and a reduction in quiet or resting behaviours (*i.e.*, weight shifting).

## Hormonal changes

**Salivary cortisol.** As expected, salivary cortisol concentration increased immediately after exercise and remained elevated one-hour post-exercise. This finding aligns with previous studies reporting increased salivary cortisol concentration in horses following various types of physical exertion, including obstacle courses [48,49], endurance and racing exercise or competition [10]. This increase is most likely to reflect the physiological demands of exercise itself, which represents a normal hormonal response to physical exercise. However, as shown in other studies, stressful contexts or constraining positions can further elevate cortisol responses [1,10,50,51]. The elevation in cortisol observed here therefore likely reflects a combination of both physiological and emotional stress.

**Adrenaline.** Our study also showed an increase in serum adrenaline concentration immediately after exercise, consistent with findings from previous studies regarding the effect of exercise on adrenaline [17]. However, we found important interindividual differences in serum adrenaline concentration increases after exercise, that could also be

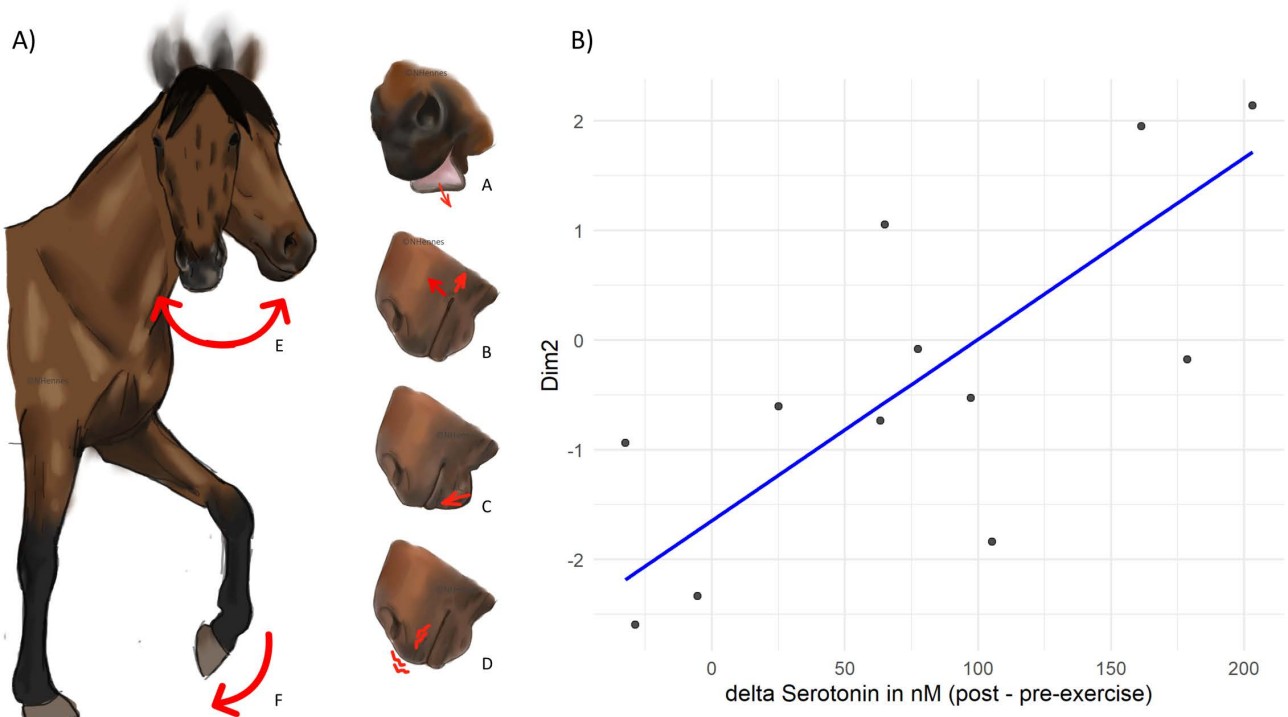

**Fig 3.** Behaviours and facial expressions associated with the second dimension ("agitation and mouth movements") of the post-exercise PCA and their relationship with serotonin variations. **A.** Behaviours and facial expressions correlated with the second dimension of the post-exercise PCA (Dim2): "agitation and mouth movements" (A: lower lip depressor (AU19), B: lip puller (AU113+AU12), C: chin raiser (AU17), D: upper lip tremor (ULT), E: head turning and F: Paw). Horses with higher increases in serotonin and adrenaline exhibited these behaviours more frequently or for longer durations. **B)** Observed values of Dim2 (scatter points) and the predicted values from the model (blue line) as a function of variations in serotonin (post minus pre-exercise), with adrenaline held constant.

**Table 6. Summary of the selected models results for Dim1 and Dim2 from the full model with cortisol, adrenaline and serotonin as a fixed effect and stable as random effect.** Model type refers to generalized linear models (GLM) with random effects for Horse and Stable.

| Fixed effects | Model type | Transformation | Chi² | Degree of freedom | p-value | Marginal R² | Conditional R² |
|---|---|---|---|---|---|---|---|
| Selected model: Calm vigilance (PC1) ~ 1 + (1|Stable) | | | | | | | |
| *Any* | glm | No | NA | NA | NA | 0 | 0.228 |
| Selected model: Agitation and mouth movements (PC2) ~ serotonin+adrenaline + (1|Stable) | | | | | | | |
| *Serotonin* | glm | No | 29.33 | 1 | <0.001 | 0.742 | 0.742 |
| *Adrenaline* | glm | No | 8.74 | 1 | <0.01 | | |

dependent on different factors such as environmental factors, age, sex or emotional stress in both humans [8] and horses [3].

**Serotonin.** For serotonin (5-HT) concentration, we observed a small but significant increase after exercise, as observed in previous studies studying the evolution of 5-HT after trekking [52], treadmill work [27] and after gallop race and exercise session [1]. However, the effect of time explained less than 5% of the variance in 5-HT concentration, while random effects were particularly strong (marginal $R^2 = 0.0413$ vs. conditional $R^2 = 0.797$), highlighting substantial inter-individual variability in the serotonergic response.

**Table 7. Summary of model estimates for "agitation and mouth movements" (Dim2) with post-minus-pre-exercise variations in adrenaline and serotonin as fixed effects. Random effects include Stable. The table presents estimates (Est/Beta), standard errors (SE), 95% confidence intervals (95% CI), z-values, and p-values. The final column reports the Chi² statistic and p-value comparing each model to the null model (i.e., models with random effects only).**

| | Est/Beta | SE | 95% CI | z-value | p | Comparison to null model |
|---|---|---|---|---|---|---|
| **Agitation and mouth movements (PC2) ~ serotonin + adrenaline + (1\|Stable)** | | | | | | |
| *Adrenaline* | 0.005 | 0.002 | [0.002;0.008] | 2.96 | <0.01 | $Chi^2 = 15.5$ |
| *Serotonin* | 0.017 | 0.003 | [0.011;0.023] | 5.42 | <0.001 | $P < 0.001$ |

For all three hormones studied, concentrations returned to baseline levels 24 hours after exercise, suggesting effective recovery from exercise-induced stress.

## Changes in behaviours and facial movements after intense exercise

Differences were effectively found in behaviours and facial movements between pre and post-exercise. Indeed, the behavioural patterns differed from a component that was associated with a range of facial movements located on the mouth as well as ears forwards. In other terms, horses had more mouth movements and spent more time with the ears forwards post-exercise than pre-exercise. This finding indicates a behavioural response to exercise, especially regarding their facial movements. The fact that these differences are for most part located on the mouth may suggest that there could be a rebound effect on the bit after exercise. Rein tension, even if not assessed for the present study, is known to be substantial in harness trotting [53], and as reins are directly attached to the bit, the mouth is likely to experience considerable pressure during an interval training exercise. It is therefore plausible that the increased frequency of mouth movements observed post-exercise reflects, at least in part, a release or rebound effect following bit-related pressure.

The longer time spent with ears forwards after exercise compared to before differed from other study [28] who observed the contrary. However, they were observing ranch horses after several hours of low to median intensity work whereas we observed horses after a higher intensity and much shorter exercise which could explain the difference observations. Indeed, whereas the study of ranch horses seemed to have observed behaviours related to physical tiredness and slight soreness after exercise, we may have captured behaviours of post-exercise arousal and neurophysiological rebound after exercise.

## Interrelations between hormonal changes, behaviours and facial movements

To explore potential links between behaviours/facial movements and hormone concentrations, we chose to use their post-exercise frequency and duration rather than the difference between pre- and post-exercise measures. This approach aimed to identify behavioural patterns that could be readily observable without requiring prior knowledge of each horse's baseline behaviour, which may not be feasible in practice. In race stables in particular, the high turnover of both horses and personnel [54,55] often makes it difficult to develop an in-depth understanding of individual behavioural baselines, a process that would be time-consuming and require specific expertise.

Post-exercise behaviours and facial movements varied partially between individuals along a dimension characterised by several mouth-related actions; including lower lip depressor (AU16), sharp lip puller and lip corner puller (AU113 + AU12), chin raiser (AU17), upper lip tremor (ULT); as well as head turning and pawing. This dimension was negatively associated with weight shifting, a behaviour previously suggested to reflect physical tiredness [28]. Interpreted as reflecting "agitation and mouth activity," this behavioural profile was linked to higher increases in serotonin and adrenaline. In other words, individuals showing the highest hormonal elevations for these two hormones tended to be more agitated, while those with lower increases appeared calmer and more prone to resting behaviours. Moreover, when we modelled this second "agitation and mouth movements" dimension using

the hormonal changes (i.e., cortisol, adrenaline, serotonin) as fixed effects, we found a positive association with post-exercise increases in both adrenaline and serotonin. Horses showing the highest increases in these two hormones, especially serotonin, also displayed more agitation-related behaviours than those with lower post-exercise hormonal changes. This link may reflect the effects of both adrenaline and serotonin on attention and alertness [56,57], thereby increasing behavioural responsiveness immediately following exercise. As hormones remained elevated directly after exercise, physiological systems were still under the influence of these changes. This ongoing activation may lead to a physiological rebound effect, whereby greater hormonal changes are associated with more pronounced behavioural and facial responses.

If we consider that higher increases in serotonin and adrenaline reflects higher exercise-induced stress, we can consider that the association of these described behaviours and facial movements could be used as indicator of exercise-induced stress during exercise. This interpretation is particularly supported by the increased frequency of pawing behaviour, which has been identified as a stress-related indicator in several studies [58,59]. Regarding mouth movements, four specific mouth movements were correlated with the axis linked with hormonal elevations. Among them, lower lip depressor has already been observed elevated during orthopaedic pain, but it co occurred with other facial expressions such as half blink [60] that we don't observed in our study. As mentioned previously, these oral movements might rather reflect a rebound effect following bit pressure during exercise. Consequently, an increased frequency of such movements could indicate stronger bit-related stimulation, and thus higher stress levels, aligning with the observed association between these movements and elevated adrenaline and serotonin levels. However, the absence of a significant link between the "agitation and mouth movements" dimension and changes in salivary cortisol suggests that these results should be interpreted with caution, as cortisol is considered a key physiological marker of stress [61]. Even if an explanation could be that the lower inter-individual's differences in salivary cortisol changes could explain this lack of association.

Additionally, mouth movements and hormones increase may be directly related. This hypothesis would be aligned with other studies showing that elevated serotonin can enhance certain types of facial muscle activity. For instance, Bershad et al. [62] reported increased zygomatic activation in response to touch following pharmacologically elevated serotonin levels in humans and another study found that serotonin can control rhythmic whiskers' movements in rodents [63]. Even if those two studies deal with central serotonin, it may be possible that peripheral serotonin have similar effects [64]. Considering this latter hypothesis, that higher serotonin levels may facilitate certain facial movements, our results may reflect an increase in mouth movements in horses with elevated serotonin following exercise. This interpretation suggests that serotonin may modulate somatic expression in the post-exercise period. In support of this idea, Trindade et al. [28] found that tired horses that exhibited frequent weight shifting also showed decreased responsiveness to tactile stimuli, such as reduced skin movements. In our study, we didn't measure skin movements as horses wore blankets but weight shifting was negatively associated with the "restlessness and mouth movements" axis, suggesting that horses showing less fatigue-related behaviour and more mouth activity may also have experienced a greater increase in serotonin. Taken together, these results suggest a possible link between serotonin reactivity and the expression of somatic or facial responses following intense exercise. Finally, the positive association between adrenaline and this behavioural dimension is consistent with the known role of adrenaline in stimulating the contractile process [3] which may contribute to the increased frequency of mouth muscle movements.

## Limitations

It should be noted that despite the significant relationships we found between certain behaviours, facial movements and changes in hormone levels (i.e., adrenaline and serotonin), these factors only explained a small proportion of the variability in post-exercise behaviours and facial movements between horses. Specifically, the first dimension of our PCA on

post-exercise behaviours and facial movements did not show any correlation with changes in these hormones, suggesting that other factors must be considered to fully explain the expression of behaviours and facial movements following high-intensity exercise. These factors may include variables such as gender, age, emotional state, training and competition experience.

Despite the limited number of individuals, this study revealed associations between post-exercise behavioural profile and hormonal changes. However, these findings represent preliminary information toward understanding this relationship. Further research is needed, particularly to investigate the influence of additional factors.

## Conclusion

This study highlights that trotting exercise induces a hormonal response characterised by a significant increase in salivary cortisol, serum adrenaline and serotonin immediately after exercise, with levels remaining elevated for at least one hour (cortisol and adrenaline), as well as changes in behaviours and facial movements. These hormonal fluctuations are associated with an increase in the expression specific facial movements and especially mouth movements.

The observed associations of hormones involved in exercise-induced stress and stress regulation with various mouth movements and agitation-related behaviours suggest that these behavioural indicators could serve as valuable tools for assessing French Standardbred serotoninergic and adrenergic responses to exercise, potentially reflecting their perception of stress during exercise. However, further studies are needed to confirm whether these associations are directly mediated by hormonal changes or result from a combination of physiological responses induced by exercise. In addition, distinguishing the stress induced by exercise from that induced by environmental factors and emotional states remains an important area for future research.

## Supporting information

**S1 File. Supplementary tables (S1_Table–S4_Table).**
(DOCX)

## Acknowledgments

The authors would like to Philippe De Deurwaerdere for performing the adrenaline assays.

## Author contributions

**Conceptualization:** Noémie Hennes, Aline Foury, Arnaud Duluard, Alice Ruet, Léa Lansade.

**Data curation:** Noémie Hennes, Léa Tutin, Aline Foury, Sylvie Vancassel.

**Formal analysis:** Noémie Hennes, Léa Tutin, Aline Foury, Sylvie Vancassel, Alice Ruet.

**Funding acquisition:** Noémie Hennes.

**Investigation:** Noémie Hennes.

**Methodology:** Noémie Hennes, Alice Ruet.

**Project administration:** Noémie Hennes.

**Supervision:** Alice Ruet, Léa Lansade.

**Visualization:** Noémie Hennes.

**Writing – original draft:** Noémie Hennes.

**Writing – review & editing:** Aline Foury, Hélène Bourguignon, Arnaud Duluard, Alice Ruet, Léa Lansade.

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
