## [Decision Letter · Decision Letter 0]

22 Aug 2025

Dear Dr. HENNES,

We look forward to receiving your revised manuscript.

Kind regards,

Metha Chanda, D.V.M.,Ph.D., DTBVM

Academic Editor

PLOS ONE

Journal Requirements:

2. To comply with PLOS ONE submissions requirements, in your Methods section, please provide additional information regarding the experiments involving animals and ensure you have included details on methods of anesthesia and/or analgesia.

3. Thank you for stating the following financial disclosure: [This work received fundings from the French National Institute for Horse and Riding (CSIFCE2022), the French association for research and technology and the French horse racing federation (CIFRE contract 2022/0468).]. 

5. Please include captions for your Supporting Information files at the end of your manuscript, and update any in-text citations to match accordingly. Please see our Supporting Information guidelines for more information: http://journals.plos.org/plosone/s/supporting-information .

Additional Editor Comments:

Dear author,

I appreciate your time and dedication in conducting this work. Based the reviewers' response, a major revision is recommended. Please address point-by-point raised by the reviewer 1 and submit it accordingly.

Best regards,

Metha Chanda

Assigned editor

Reviewers' comments:

Reviewer's Responses to Questions

**Comments to the Author**

1. Is the manuscript technically sound, and do the data support the conclusions?

Reviewer #1: No

Reviewer #2: Yes

2. Has the statistical analysis been performed appropriately and rigorously?

Reviewer #1: Yes

Reviewer #2: Yes

3. Have the authors made all data underlying the findings in their manuscript fully available?

Reviewer #1: Yes

Reviewer #2: Yes

4. Is the manuscript presented in an intelligible fashion and written in standard English?

Reviewer #1: Yes

Reviewer #2: Yes

Reviewer #1: Overall this is quite a well conducted study measuring the endocrine response and behaviour post interval training in horses.

However, the problem is the manuscript is written as if there is causality between the physiological effects of exercise and the post exercise behaviour without considering the confounding effect of 1. cross tying horses and 2. restricting movement and rugging hot horses after intense exercise.

The hypotheses could equally be - does cross tying horses cause increase oral related, weight shifting and stress behaviours - or if the horses were also cross tied before exercise (just not stated) - does cross tying hot horses post intense exercise increase oral, shifting, and stress related behaviours.

Although the precise duration and intensity of exercise are not well described, physiologically, horses will have accumulated a large heat burden, continue to sweat and behaviourally would be inclined to want to walk and roll - which is restricted by cross tying them.

The endocrine responses to exercise are not unusual and don't in themselves reflect a negative "stress" perception by the horse, but a normal physiological response to exercise.

The discussion concludes that the behaviour is exercise-induced stress, but this cannot be concluded without the removal of the environmental stressors post exercise e.g. a control group that were cooled down properly and allowed access to a sand roll or unrestricted movement.

The fact that the horses were more aroused post exercise is unsurprising and the degree of arousal reflected in ADR and 5HT being partly related to the increased frequency of behaviours is also expected.

Title/Abstract/Introduction:

Trotting exercise is non specific and presumably refer to interval training - use the most specific term throughout

Speculation about the bit pressure in the abstract appears unfounded by the dataset

Have the authors excluded other causes for agitation post exercise e.g. could be associated with sweating, cooling down etc.

Some grammatical errors e.g. "chronical" instead of chronic, "had a shower", Line 164 "Dosages" observers were validated to use...

Methods:

Missing details relevant to study

1. duration of turnout in hours (versus frequency per week)

2. Training status of the horses i.e. were they race fit or still in training

3. wrt exercise load in the interval session/Duration of intervals (distance or time) - average does not indicate how long the horses spend at maximal or high intensities versus recovery

4. Were horses cooled down? Were they scraped as well as rinsed? Were they allowed to roll e.g. in a sand roll following exercise?

5. When were the behaviours and facial movements recorded? Also being cross tied (line 209) is different from being loose in the stall as implied earlier (Line 128 )- clarify

Discussion -

Line 414 - 415 - in order to conclude as per the Rollkur argument would need to compare cortisol response post interval training in your horses and previous research. Indeed the Rollkur argument agrees with the alternate hypothesis that restriction of hot horses with cross tying port interval training is the stressor - but only if the post exercise cortisol is greater than horses exercise of equivalent intensity and workload.

Lines 431-434 unreferenced and not clearly related to the study.

The hypothesis of the rebound effect from the bit is unfounded from the data in the study. Did the horses in this study have evidence of mouth lesions? Was the degree of rein tension even observed?

Reviewer #2: Really interesting idea to study, relative perceived effort (RPE) is widely used in sports science as a metric for measuring training load, progress and managing recovery. Correctly applied and interpreted this has beneficial impacts on athlete welfare.

For the horse where it is not possible to use self reflections to gauge the mood-related, emotional, or stress perception of components of the stress response I think this multimodal approach takes into consideration the breadth of the stress response to exercise nicely.

I would love to see a replication of this test to see to what extent individual variation persists, and how these metrics might change through the duration of a training program i.e. is there a period of acclimatisation.

Line 53 - a minor typo - ‘Chronical’ - I think this should say "Chronically"

The limitations and cautions for the results are well-discussed. Overall a really nice study, and one which is really positive for trainers and owners — to know that you can spend time observing your horse (with an educated eye or ethogram guide) to assess their response to training is a really important message at this time when welfare in horse sport is so crucial.

**Do you want your identity to be public for this peer review?** For information about this choice, including consent withdrawal, please see our Privacy Policy

Reviewer #1: No

Reviewer #2: No

---

## [Author Response · Author response to Decision Letter 1]

5 Oct 2025

A detailed response to all reviewer and editor comments is provided in the uploaded file ‘Response to Reviewers.docx’. All requested revisions have been made accordingly.

---

## [Decision Letter · Decision Letter 1]

19 Oct 2025

Exploring the association between stress-related hormonal changes, behaviours and facial movements after a trotting exercise in French Standardbred

PONE-D-25-35600R1

Dear Dr. HENNES,

We’re pleased to inform you that your manuscript has been judged scientifically suitable for publication and will be formally accepted for publication once it meets all outstanding technical requirements.

Kind regards,

Metha Chanda, D.V.M.,Ph.D., DTBVM

Academic Editor

PLOS ONE

Additional Editor Comments (optional):

Dear Author,

I appreciate your time and effort to revise this manuscript based on two reviewers' comment, after careful evaluation, I recommend acceptance of the manuscript in its current form.

Best regards,

Metha Chanda

Reviewers' comments:

Reviewer's Responses to Questions

**Comments to the Author**

Reviewer #1: All comments have been addressed

2. Is the manuscript technically sound, and do the data support the conclusions?

Reviewer #1: Yes

3. Has the statistical analysis been performed appropriately and rigorously?

Reviewer #1: Yes

4. Have the authors made all data underlying the findings in their manuscript fully available?

Reviewer #1: Yes

5. Is the manuscript presented in an intelligible fashion and written in standard English?

Reviewer #1: Yes

Reviewer #1: "Showered" should be changed to "rinsed" and the discussion of oral focused behaviour is still focused on the bit during exercise rather than cross tying/restricting movement in a hot aroused horse following exercise but otherwise the authors have thought about my comments carefully and addressed them to my satisfaction.

**Do you want your identity to be public for this peer review?** For information about this choice, including consent withdrawal, please see our Privacy Policy

Reviewer #1: No

---

## [Editor Report · Acceptance letter]

PONE-D-25-35600R1

PLOS ONE

Dear Dr. HENNES,

I'm pleased to inform you that your manuscript has been deemed suitable for publication in PLOS ONE. Congratulations! Your manuscript is now being handed over to our production team.

Kind regards,

on behalf of

Associate Professor Metha Chanda

Academic Editor

PLOS ONE